Subject Area:
biophysics

Keywords:
photosynthesis, non-photochemical quenching, energy-dependent quenching, snapshot spectroscopy, multiscale models, excitation energy transfer

Author for correspondence:
Graham R. Fleming
e-mail: grfleming@lbl.gov

# Models and mechanisms of the rapidly reversible regulation of photosynthetic light harvesting

Doran I. G. Bennett[1], Kapil Amarnath[2], Soomin Park[3,4], Collin J. Steen[3,4], Jonathan M. Morris[3,4] and Graham R. Fleming[3,4]

[1]Department of Chemistry and Chemical Biology, Harvard University, Cambridge, MA 02138, USA
[2]Department of Physics, University of California San Diego, La Jolla, CA 92093, USA
[3]Department of Chemistry, University of California, Berkeley, CA 94720, USA
[4]Molecular Biophysics and Integrated Bioimaging Division, Lawrence Berkeley National Labs, Berkeley, CA 94720, USA

 DIGB, 0000-0001-8322-7371; SP, 0000-0001-6787-2098; CJS, 0000-0002-7029-2892; GRF, 0000-0003-0847-1838

The rapid response of photosynthetic organisms to fluctuations in ambient light intensity is incompletely understood at both the molecular and membrane levels. In this review, we describe research from our group over a 10-year period aimed at identifying the photophysical mechanisms used by plants, algae and mosses to control the efficiency of light harvesting by photosystem II on the seconds-to-minutes time scale. To complement the spectroscopic data, we describe three models capable of describing the measured response at a quantitative level. The review attempts to provide an integrated view that has emerged from our work, and briefly looks forward to future experimental and modelling efforts that will refine and expand our understanding of a process that significantly influences crop yields.

## 1. Introduction

Plants and photosynthetic algae in natural environments experience sunlight intensities sufficient to damage the light-harvesting apparatus. When excess photons are absorbed by photosystem II the formation of reactive oxygen species, such as singlet oxygen, can cause inactivation of photosynthetic proteins [1,2]. Green plants and algae have a suite of regulatory mechanisms that are often quantified from their effect on chlorophyll fluorescence as 'non-photochemical quenching' (NPQ) and which act on various time scales to dissipate the surplus excitations [3–5]. The most rapid response, called energy-dependent quenching (qE) [3,6,7], is especially important in the response of photosynthetic organisms to naturally fluctuating light [8]. Zhu *et al.* [9] estimate that crop yields could be increased by as much as 30% by optimizing the NPQ response. Indeed, by overexpressing three genes known to be involved in qE, Kromdijk *et al.* demonstrated a 15% increase in yield in tobacco plants over a growing season [10]. These results make a clear case for developing a quantitative molecular-level understanding of qE that would enable the rational design of crops with further increased yield.

A model capable of quantitatively predicting the influence of qE on the kinetics of the light reactions in the presence of genetic and environmental perturbations could subsequently be incorporated into larger scale models of photosynthesis and crop yield. The path to a complete understanding of qE requires the development of both quantitative experimental readouts and multiscale modelling approaches capable of treating this complex process. In this review, we focus on our efforts to develop models and measurements of qE, much of which has involved collaboration with K. K. Niyogi and R. Bassi. At

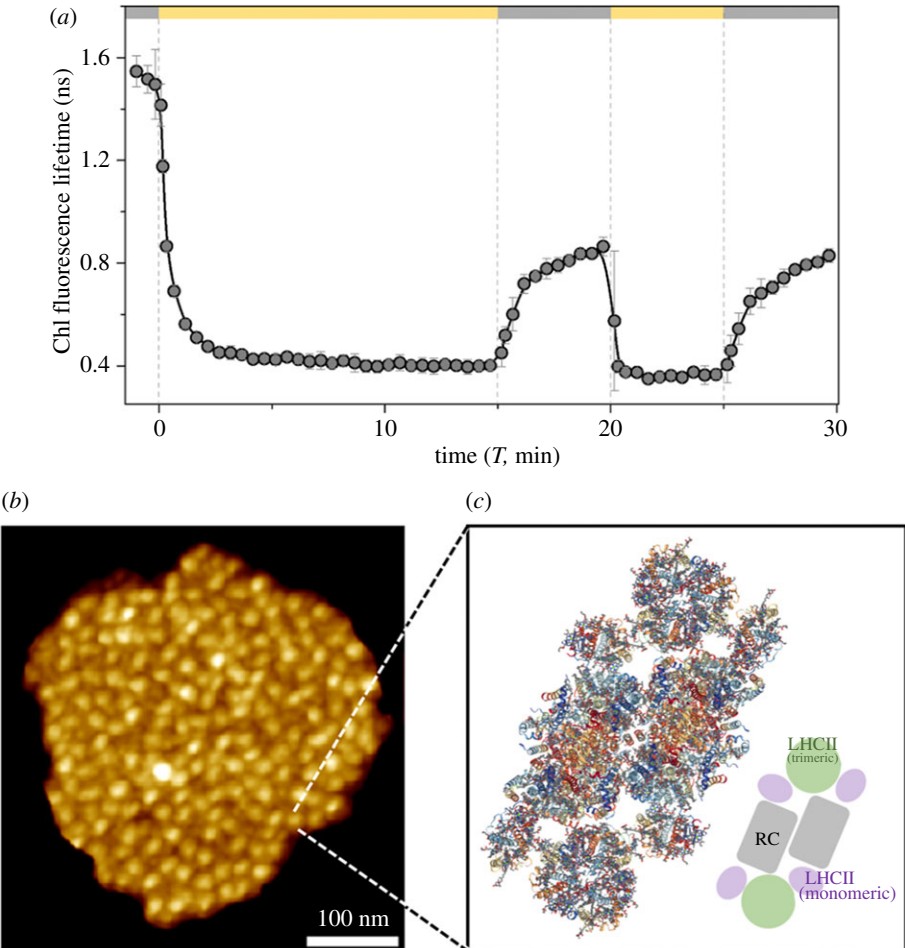

**Figure 1.** (a) Average Chl fluorescence lifetimes ($\tau_{average}$) of wild-type *Arabidopsis* leaves in response to high-light and dark exposures. Each data point is presented as average $\pm$ s.d. ($n = 3$). Detailed information on the measurements and lifetime calculation are described in the 'Snapshot measurements' section. (b) Atomic force microscopy micrographs of the grana membrane showing the spatial distribution of PSII. Reproduced from fig. 3D in [11]. (c) Structure of the spinach C2S2-type PSII-LHCII supercomplex (RCSB PDB ID: 3JCU) (structure obtained from [12]).

the end of the review, we will look back at the different approaches presented, as well as the broader literature of qE, to suggest how to integrate the models into a unified picture.

The phenomenon of qE is often measured as a reduction in the fluorescence yield following acclimation to bright light, as shown in figure 1a. A leaf was excised from an *Arabidopsis thaliana* plant and the petiole was immersed in water. The leaf was then first exposed to darkness for several minutes, before being exposed to bright light for several minutes and then returned to darkness. During this dark–light–dark cycle, the chlorophyll fluorescence lifetime was measured every 10–30 s. The average lifetime decreases, indicating fluorescence quenching, as soon as the leaf is exposed to the light. When the leaf is subsequently exposed to darkness qE turns off, but at a slower rate than the turn on. The ability of a leaf to perform this rapidly reversible fluorescence quenching, or 'qE', is correlated with higher fitness in the field [8].

The measured chlorophyll fluorescence in figure 1a arises from photosystem II, which is found in the thylakoid membrane. The photosynthetic membrane is densely packed with pigment–protein complexes which can constitute as much as 80% of the membrane surface area [13–15]. Figure 1b shows an atomic force micrograph of a grana membrane. The white spots are the photosystem II (PSII) supercomplex shown in more detail in figure 1c. Not resolved in figure 1b is the intervening collection of antenna pigment–

protein complexes between the supercomplexes, which contain the bulk of the chlorophyll (Chl) in the thylakoid membrane. These antenna complexes consist mostly of trimers of light-harvesting complex II (LHCII), each monomer of which binds 14 chlorophyll molecules (eight Chl *a*, six Chl *b*) and four carotenoid (Car) molecules (two lutein, one neoxanthin and one xanthophyll cycle carotenoid) [16–18]. Monomeric light-harvesting complexes (CP24, CP26 and CP29 in green plants) generally constitute components of the supercomplex which is itself dimeric [19].

The basic picture of PSII light harvesting is as follows: absorption of a photon of sunlight by a pigment in an antenna complex results in a nascent excitation that is ultimately transferred over tens of nanometres to the special pair of chlorophylls in the reaction centre, where charge separation converts the excitation into chemical energy [20]. When light levels are very low the capture of absorbed photon energy for photosynthetic charge separation is highly efficient. In excess light, reaction centres no longer act as efficient quenching sites (they are referred to as 'closed'), thereby increasing the probability of Chl triplet formation. Spin allowed energy transfer between excited triplet chlorophyll and ground state (triplet) oxygen results in the formation of highly reactive excited singlet oxygen [21,22]. To minimize the possibility of oxidative damage, plants and algae create alternative quenching sites to the reaction centres, and the fluorescence lifetime and yield are reduced

on a time scale of seconds to minutes. When light levels drop, these additional quenching sites are disabled on a much slower time scale (a few tens of minutes).

The primary difficulty in discussing and modelling qE is that it is highly multiscale and multidisciplinary. Though the qualitative picture was established 30 years ago, many of the individual processes involved constitute outstanding challenges in their respective fields, which span from the study of excited state dynamics to biochemical mechanisms. On the molecular scale (Å–nm) within a pigment–protein complex, what is the physical mechanism of quenching and how does excitation transport compete with this quenching process? On the membrane scale (10s of nm–μm), how does quenching affect transport across the membrane to reaction centres? While the photophysics of light harvesting takes place on the fs–ns time scales, the time scales of biochemistry (ms–s) dictate the time scale of the activation of qE in response to high light. How does this (de)activation take place? Lastly, qE is triggered by the pH gradient across the thylakoid membrane, which reflects the overall output of the light reactions [5]. The extent of qE affects the photochemical yield at the reaction centres, which affects the pH gradient. Thus, to model qE in the context of the light reactions requires embedding the photophysical and biochemical understanding of qE into a chemical network spanning all of the light reactions! The difficulty of this overall challenge has forced us to isolate particular parts of the problem with the goal of eventually unifying them into a single model that could be incorporated in a larger-scale model of crop yield.

Efforts in our group to achieve a quantitative understanding of qE have progressed along two lines: (1) the development of 'snapshot' spectroscopic tools that can directly observe photophysics *in vivo* on the biochemical time scales of qE (de)activation; and (2) multiscale models that can integrate data from many sources and scales to start to produce a unified picture.

The need for measuring the photophysical processes underlying qE which occur on the ps–ns time scale in 'snapshots' on biochemical time scales of activation (s–min) arose because of the disagreement over which photophysical signals are physiologically relevant. Despite the much greater tractability of measuring the time-resolved spectroscopy of pigment–protein complexes *in vitro* versus *in vivo*, the photophysical properties of pigment–protein complexes depend exquisitely on their environment [23–25]. Thus, *in vivo* spectroscopic data, though it is significantly more coarse, is needed to properly assess the physiological relevance of *in vitro* data. The second benefit of 'snapshot' spectroscopic tools is the ability to correlate photophysical signals with other biochemical data in the same sample. We have used such correlations in conjunction with genetic mutants of qE to develop and test mechanistic hypotheses on the quenching mechanism of qE and the biochemical activation of qE in plants.

The typical experimental readouts of qE, such as chlorophyll fluorescence, are relatively coarse measurements and their simple form belie the myriad kinetic processes underlying them. On the femto-to-nanosecond time scale, a kinetic model of PSII light harvesting consists of the thousands of excitation transport rates between pigments, the rates of charge separation in the reaction centre, and the rates of quenching by qE, fluorescence and non-radiative decay. As nearly all of these parameters cannot be directly measured *in vivo*, developing an accurate model requires a 'bottom-up' approach in which the established theory of energy transfer is parametrized using the large collection of available data, such as X-ray and cryo-EM structures of the antenna and supercomplexes [12,26–28], and spectroscopic data from both isolated complexes [29–35] and intact membranes [36–39]. Once such a model is constructed, both a multiscale photophysical understanding of PSII light harvesting in the presence of quenching and rigorous coarse-graining become feasible [40]. On a longer time scale of seconds to minutes, at each time point the state of the PSII light-harvesting apparatus depends on the biochemical reactions, such as those that determine the activation of qE. Such a kinetic model must incorporate measurements and understanding from a wide range of sources to be correctly parametrized and structured. Once such a kinetic model is in place, however, some idea of the key biochemical determinants of qE activation becomes possible.

The review is organized as follows: in the first half, we discuss the development of a phenomenological approach to studying qE using the combination of 'snapshot' spectroscopies and genetic mutants of qE. The second half we discuss the development of two multiscale models: the first involving entirely the light-harvesting component of qE and the second a kinetic model of all the light reactions including qE. The review concludes with an overview of what has been learned from this work, considers the role of NPQ on overall photosynthetic productivity, and sketches some topics for future research directions.

One substantial challenge to connecting putative mechanisms and spectroscopic signals, however, is the structural disorder of the thylakoid membrane. The thylakoid membrane is often in a 'mixed' configuration (figure 1b)—though more crystalline arrangements have also been found [41]—where there can be substantial differences in nearest-neighbour distances and the relative ratio of different proteins across regions of the membrane. The resulting measurements are then averaged over the disordered membrane which can result in the loss of spectroscopic signatures of specific mechanisms. In the measurements presented below, we do not consider the details of how excitation moves through the membrane in order to reach a quenching site and, as a result, our discussion is relatively insensitive to the specifics of the membrane organization. As more detailed measurements become available, however, analysing the spectroscopic measurements will require multiscale modelling approaches, akin to what we will discuss in the section 'Multiscale photophysical model of NPQ'.

# 2. Spectroscopic probes of excitation quenching mechanisms

There are two distinct senses in which we need to understand the mechanism of excitation quenching: what is the photophysical process that dissipates excitation energy? And what molecular processes control the activation and de-activation of the non-photochemical quenching sites?

There seem to be only a small number of possible quenching processes consistent with available evidence: excited chlorophyll (Chl) may transfer its excitation energy to another molecule whose excited singlet state has a naturally

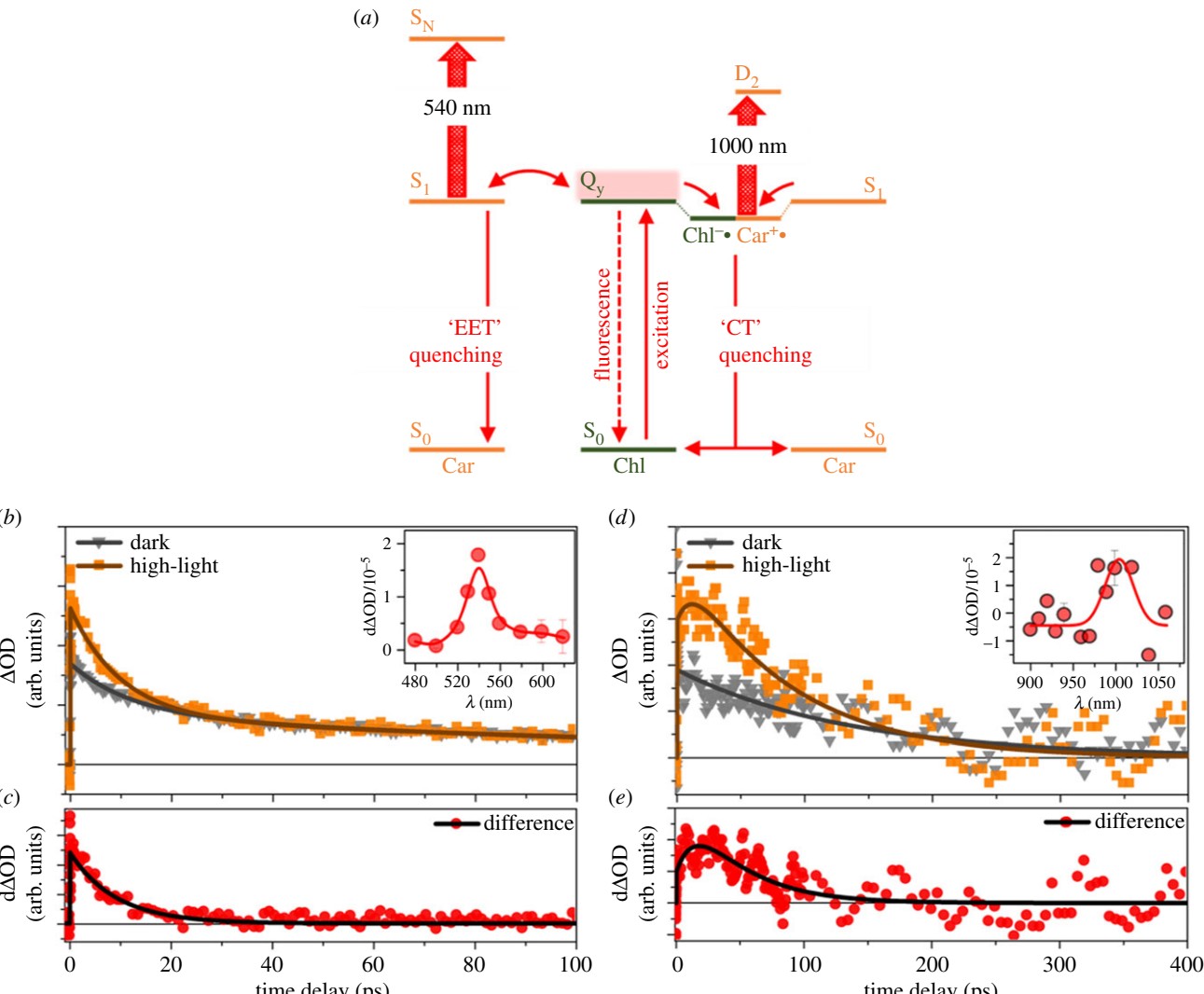

**Figure 2.** (*a*) Schematic for the Chl–Car excitation energy transfer (EET) and Chl–Car CT quenching processes. (*b*–*e*) TA kinetic profiles for spinach thylakoid membranes after Chl excitation at 650 nm. (*b*) Profiles probed at 540 nm under dark (grey, down triangle) and high-light (orange, rectangle) conditions. The high-light samples were illuminated with 850 µmol photons m$^{-2}$ s$^{-1}$ for about 15 min prior to measurement. The inset graph shows the difference TA spectrum reconstructed by subtracting the dark signal from the high-light signal at 1 ps. (*c*) Difference between high-light and scaled dark kinetic profiles measured at 540 nm. The lifetime obtained from fitting with a single exponential decay is 7.81 ± 0.83 ps. (*d*) Profiles probed at 1000 nm under dark (grey, down triangle) and high-light (orange, rectangle) conditions. The inset graph reconstructed by difference at 20 ps. (*e*) Difference between high-light and dark kinetic profiles measured at 1000 nm, which is fitted with rise (15.4 ps) and decay (40 ps) components. Panels (*b,c*) were reprinted (adapted) with permission from [39]. Copyright © 2018 American Chemical Society. Panels (*d,e*) were reprinted (adapted) with permission from [38]. Copyright © 2017 American Chemical Society.

short lifetime [36,42,43]. Or charge transfer (CT) may occur, followed by charge recombination to the ground state. The former mechanism is referred to as excitation energy transfer (EET) quenching, and the latter one as CT quenching. CT quenching may involve a pair of chlorophylls (Chl–Chl CT) [34,44] or a chlorophyll and a carotenoid (Chl–Car CT) [31,37]. Given the biochemical, genetic and spectroscopic evidence for the involvement of zeaxanthin (Zea), many authors have considered the interaction of zeaxanthin (and other xanthophylls, particularly lutein), with chlorophyll. Electronic structure calculations by Dreuw *et al.* [45] showed that, in the case of Chl-xanthophyll mixing, strong CT interaction occurs and at separations of approximately 5.5 Å the lowest-energy excited state is not a neutral exciton but a state involving complete transfer of one electron from the xanthophyll to the chlorophyll. The calculations further show that zeaxanthin is the most prone to forming a CT state, followed by antheraxanthin, with violaxanthin requiring the greatest proximity to give a low-energy CT state [46].

These proposals for the photophysical process of excitation dissipation suggest that activating quenching might only require a small change in pigment separation/conformation. Several chemical signals have been suggested to induce the requisite structural changes, including an LHCII aggregation-induced model [6,44,47] and a pH gradient induced conformational change in antenna proteins [35]. Addressing these putative mechanisms for quenching activation requires connecting the spectroscopic probes of the ultrafast excitation dissipation with the biochemical measurements associated with the activation/de-activation dynamics.

## 2.1. Steady-state measurements

Transient absorption (TA) measurements of thylakoid membranes in a known steady state (i.e. 'light acclimated' or 'dark acclimated') provide evidence for quenching by both the Car $S_1$ state (EET quenching) and Car$^{\bullet+}$ states (CT quenching; figure 2*a*). We begin by considering the evidence

royalsocietypublishing.org/journal/rsob    Open Biol. 9: 190043

for EET quenching. After Chl excitation (650 nm), an excited-state absorption is found at 540 nm (assigned to the Car $S_1 \rightarrow S_N$ transition, figure 2b) with a short lifetime component (7.81 ps) that is consistent with previously reported Zea $S_1$ state (9 ps compared to approximately 14 ps for Lut $S_1$ state) [48]. The difference between the high-light adapted and low-light adapted signals measured at these wavelengths (d$\Delta$OD), shown in figure 2c, is a higher amplitude of the short lifetime component, suggesting an increased population of short-lived Car $S_1$ states. Plotting this difference between TA signals in high- and low-light acclimated thylakoids at 1 ps results in a spectrum (figure 2b, inset) that strongly resembles the $S_1$–$S_N$ absorption spectrum of Zea [48].

Carotenoid to chlorophyll CT quenching is supported by the appearance of an excited-state absorption signal around 1000 nm, consistent with the $D_0$–$D_2$ transition of $Zea^{\bullet+}$. Dark-acclimated thylakoids excited at 650 nm show a decaying excited-state absorption signal at 1000 nm with no appreciable rise time (figure 2d). Following light acclimation, however, both a 15 ps rise component and a 40 ps decay component appears. These new time components are visualized by plotting the difference between the high-light adapted and low-light acclimated signals measured at these wavelengths (d$\Delta$OD), shown in figure 2e. The difference spectrum between high- and low-light acclimated signals at 20 ps (figure 2d, inset) is consistent with the $D_0$–$D_2$ transition of $Zea^{\bullet+}$.

Although this steady-state TA data provide evidence on the active Chl–Car EET and CT mechanisms after high-light exposure (30 min), they face substantial limitations in unveiling the mechanisms of qE. Most obviously, steady-state measurements are not capable of correlating changes in biochemistry (e.g. mutations) with the activation/de-activation dynamics of qE response measured spectroscopically. As a result, we need new spectroscopic probes to correlate biochemical changes to the time dependence of the qE response.

## 2.2. Snapshot measurements

Snapshot spectroscopies are a class of measurements which aim to collect spectroscopic data during a biochemical response of the photosynthetic apparatus to a perturbation. Pulse-amplitude modulated (PAM) fluorescence is a classic example of a snapshot measurement [49]. In one common implementation, a dark-acclimated leaf is exposed to an actinic light source and the fluorescence amplitude is measured as a function of time. Periodic saturating flashes are used to close all RCs and the resulting fluorescence amplitudes report on the quenching in the absence of open RCs [50]. Our group has developed two new snapshot techniques; the first measures fluorescence lifetimes during the course of acclimation (called 'fluorescence lifetime snapshots') [51]. More recently, we have developed a technique, known as 'snapshot TA' [38,39], that is capable of resolving the ultrafast signals of excitation dissipation as a function of the slower biochemical dynamics that are responsible for quenching activation and de-activation.

Below, we highlight how snapshot spectroscopies can be used to address questions such as: How fast are the Chl–Zea EET and CT pathways activated in the response of plant photosynthetic membranes to high light? Are these processes reversible upon returning to the dark? The answers to these questions are important constraints for models of the molecular mechanisms that activate/de-activate quenching in response to changes in light intensities.

By resolving the fluorescence 'lifetime', rather than only the 'amplitude', the fluorescence lifetime snapshot technique provides an additional dimension of information during light exposure. Fluorescence decay profiles were acquired in 0.2 s 'snapshot' windows every 10–30 s. During the 'snapshot', reaction centres are saturated and the resulting fluorescence lifetimes are measured when all reaction centres are closed. Each decay curve was fit to a sum of two or three exponential decay components. The average fluorescence lifetime values ($\tau_{\text{average}}$) were calculated using the following equation:

$$\tau_{\text{average}} = \frac{\sum_i A_i \tau_i}{\sum_i A_i}. \tag{2.1}$$

Additionally, based on the $\tau_{\text{average}}$ values, a lifetime-based NPQ$\tau$ parameter can be suggested, which is analogous to the conventional NPQ value (NPQ $= (F_m - F_m')/F_m'$) from PAM fluorescence measurements:

$$\text{NPQ}_\tau = \frac{\tau_{\text{average}}^{\text{dark}} - \tau_{\text{average}}^{\text{light}}}{\tau_{\text{average}}^{\text{light}}}. \tag{2.2}$$

Figure 3a shows fluorescence lifetime data acquired in 0.2 s 'snapshots' every 10–30 s for spinach thylakoid membranes (red diamonds, $\tau_{\text{average}}$; blue squares, NPQ$\tau$). A fast activation (less than 3 min) of NPQ upon exposure to bright light is paired with a fast but partial de-activation upon return to dark conditions.

TA snapshots provide evidence for the rapid activation of both Chl–Car EET and CT quenching mechanisms in response to high light as well as de-activation upon a return to dark conditions. Figure 3b plots the excited-state absorption signals measured at 540 nm (associated with Car S1/EET quenching, solid line) and 1000 nm (associated with $Car^{\bullet+}$/CT quenching, dashed line) during high-light exposure. Both ESA signals show a rapid rise and reach a maximum level within 3 min of high-light exposure. The ESA signals normalized to the concentration of Zea pigments are well correlated with the evolution of lumenal pH which is calculated based on the kinetic model developed by Zaks et al. [52] (figure 3c). This observation suggests that the initial spike of $\Delta$pH is responsible for the rapid formation of EET and CT quenching sites during the early (less than 3 min) stage of high-light exposure. In the subsequent period of dark exposure, the Car $S_1$ signal (EET) rapidly disappears, while the $Car^{\bullet+}$ signal (CT) does not completely disappear within 5 min.

To investigate how the EET and CT are triggered in plants, two important processes were inhibited by means of chemical (1,4-dithiothreitol (DTT) and 3,3′-dithiobis(sulfosuccinimidyl propionate) (DTSSP) assays. DTT is well known to inhibit the enzyme violaxanthin de-epoxidase (VDE) and thereby the accumulation of Zea, while DTSSP is a cross-linker preventing reorganization of membrane proteins which could be catalysed by the $\Delta$pH-sensing photosystem II subunit S (PsbS) protein [53]. Interestingly, the Car $S_1$ signal (EET) was eliminated by DTT treatment [39], while the $Car^{\bullet+}$ signal (CT) completely disappeared upon cross-linking by DTSSP [38]. Although both the Car $S_1$ and $Car^{\bullet+}$ signals partially decrease following treatment by

royalsocietypublishing.org/journal/rsob    Open Biol. 9: 190043

**Figure 3.** (a) Evolution of average Chl fluorescence lifetimes ($\tau_{average}$) in response to high-light and dark exposure and the calculated NPQ$\tau$ values (see text) at each time point. Panel (a) was reprinted (adapted) with permission from [38]. Copyright © 2017 American Chemical Society. (b) Snapshot TA data of Car $S_1$ and Car$^{\bullet+}$ ESA after Chl excitation (650 nm) in spinach thylakoid membranes. The absorption for Car $S_1$ and Car$^{\bullet+}$ were probed at 540 nm and 1000 nm, respectively, following excitation at 650 nm. The vertical lines (grey, dashed) and bars at the top of the figures indicate the time sequence of actinic light on (yellow) and off (dark grey). The solid black line and dotted line are the smoothed line from the data points corresponding to Car $S_1$ ESA and Car$^{\bullet+}$ absorption (dotted line) from high-light-exposed spinach thylakoids, respectively. Note that the Zea pigment seems to be predominantly responsible for Car S1 and Car$^{\bullet+}$ which have peak absorption at 540 and 1000 nm, respectively (see main text and the inset figures in figure 2b,d). (c) Evolution of normalized Car $S_1$ absorption (Car $S_1$ ESA/[Zea]) (black squares), normalized Car$^{\bullet+}$ absorption (Car$^{\bullet+}$ ESA/[Zea]) (grey circles), and the calculated lumenal [H$^+$] (blue line) in response to high-light/dark exposure. [Zea] was determined by time-resolved HPLC measurements, while the lumenal [H$^+$] is based on the kinetic model described by Zaks et al. [52]. Note that there is considerable uncertainty in the lumenal [H$^+$] during the second high-light exposure period as the model was devised for completely dark-adapted systems. (d) Proposed scheme for the triggering system of the EET and CT quenching mechanisms involved in qE. Regarding the involvement of PsbS and Zea, essential steps are denoted by solid arrows, and non-essential but influential steps are denoted by the dashed arrow. VDE* and PsbS* represent activated proteins by $\Delta$pH and protonation, respectively. In the very initial stages of high-light exposure, CT quenching appears to depend on the small pool of Zea (or antheraxanthin) that is present in the dark. Panel (a) was reprinted (adapted) with permission from [38]. Copyright © 2017 American Chemical Society. Panels (b–d) were reprinted (adapted) with permission from [39]. Copyright © 2018 American Chemical Society.

DTSSP and DTT, respectively, the signals are not removed completely.

We suggest that the pH gradient formed in the presence of high light activates two parallel qE pathways, summarized in figure 3d. For one pathway, the enzyme VDE acts as a pH sensor and converts violaxanthin to zeaxanthin, which is an efficient EET quencher. For the second pathway, a $\Delta$pH-sensing integral membrane protein (photosystem II subunit S, PsbS) [53–55] activates a Chl–Car CT quenching mechanism in the light-harvesting proteins (e.g. LHCII). Given that DTT is specific for Zea, it is likely that the carotenoid involved in quenching chlorophyll excitation is Zea. Additional studies have identified the formation of Chl–

Zea EET and CT quenching sites in live cells of the unicellular algae *Nannochloropsis oceanica* [56]. It was found that both a $\Delta$pH-sensing protein (LHCX1) and the VDE enzyme are essential for both the Zea $S_1$ (EET) and Zea$^{\bullet+}$ (CT) signals in *N. oceanica*.

The combination of our TA data including the $S_1$ lifetime (figure 2c) and $S_1$–$S_N$ spectrum (figure 2b, inset) suggest that Zea should be considered as the most important Car quencher for Chl* quenching through both EET and CT mechanisms. The inhibition of VDE by DTT chemical treatment [38] substantially decreases the overall NPQ capabilities, which also suggests the important role of the Zea pigment in creating quenching sites for excited Chl. We

are not yet able to quantify the absolute contribution of each mechanism and so cannot rule out the other mechanisms mentioned above. One reason for this is that it has not yet proven possible to carry out snapshot TA on thylakoids of *Arabidopsis* plants where multiple qE mutants are available. This situation is likely to be remedied in the near future via mutants of *N. benthamiana*, which do give clear TA signals from thylakoids.

It is clearly important to estimate if energy and CT can account for all of qE. However, at least two factors complicate such an estimate. First, the influence of exciton annihilation in the snapshot TA measurements needs to be accurately modelled via a multiscale approach. Second, accurate values for the extinction coefficients of Zea $S_1$ and Zea$^{\bullet+}$ *in vivo* are required. Particularly in the latter case, there is a great deal of uncertainty about the value, in part because of proton loss from the radical [38,57].

# 3. Modelling the biochemistry of non-photochemical quenching

Biochemical models of non-photochemical quenching, particularly the rapidly reversible component qE, address how different biological systems couple to generate the emergent photoprotective behaviour. As we have seen above, there is a complex interplay between the activation and de-activation of different components of the qE and qZ during response to variable light conditions [58,59]. One approach to disentangling the underlying dynamics is to take the view of an engineer: construct a model with various feedback loops, time scales of response and connectivities between more or less 'black boxes' of sensing and quenching components. We will refer to such models as engineering models and illustrate them via the model developed by Zaks *et al.* [52]. An engineering approach requires parametrizing a large set of differential equations—which can limit its utility for analysing large quantities of experimental data. Alternatively, one can take a more coarse-grained kinetics approach which fits a small number of phenomenological equations (extracted from the engineering model) to data from different plant mutants. We will briefly describe such an approach by Leuenberger *et al.* [60] using various xanthophyll cycle mutants of *A. thaliana*. Both of these approaches have their merits and deciding between them requires careful consideration of the research goals.

## 3.1. An 'engineering' model of qE biochemistry

A schematic description of the model developed by Zaks *et al.* [52] to describe the qE component of NPQ is shown in figure 4. The ability of the photosynthetic electron transfer system (the 'plant' in figure 4) to use the energy contained in the excited chlorophylls (Chls) ('input', green box) determines the requirement for qE. The model assumes qE is the 'controller' (orange box) which is triggered by the lumen pH (i.e. the $\Delta$pH across the thylakoid membrane) (light blue box). The central box lists the pathways and rates of excited Chl relaxation used in the model. The lower box shows that the components involved in the activation of qE (orange box) are the protonated PsbS protein and a de-epoxidized xanthophyll. Both of these components are triggered by the lumen pH but with independently fit $pK_a$s

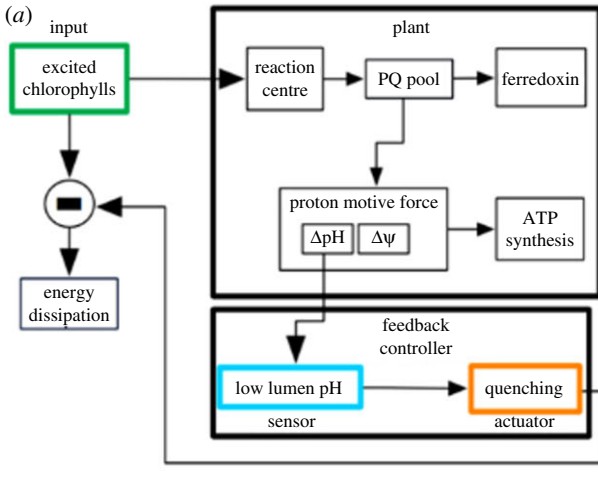

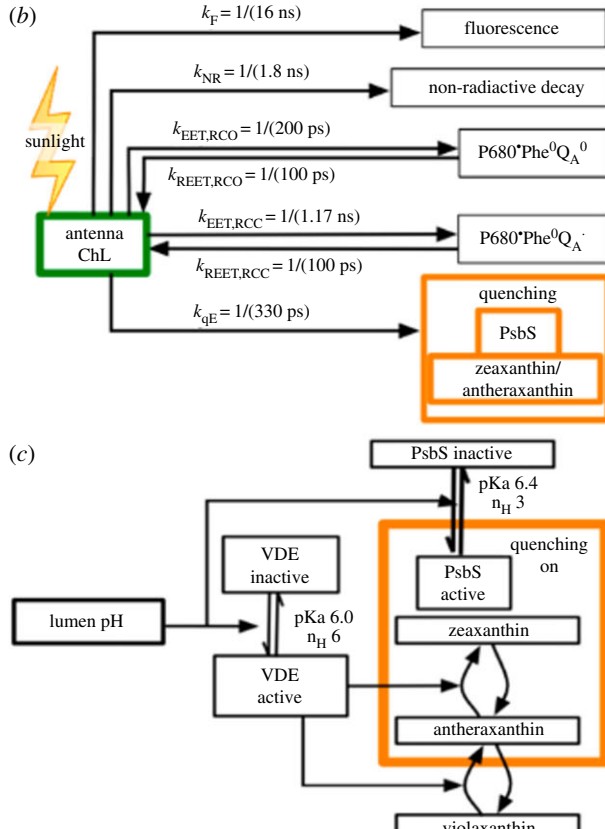

**Figure 4.** (*a*) Schematic of the system that activates and is affected by qE. qE regulates the concentrations of excited chlorophylls in the PSII antenna, which is directly affected by the light intensity. The ability of the photosynthetic electron transfer system (the 'plant') to use the energy contained in the excited chlorophylls ('input', green box) determines the requirement for qE. We consider qE to be the 'controller' (orange box) that is triggered by the lumen pH (light blue box).The lumen pH is a component of the pmf driving ATP synthesis. (*b*) Modelled pathways and rates for quenching of chlorophyll fluorescence (green box) in PSII. Quenching by qE is shown in orange. (*c*) Components involved in the activation of qE (orange box) are a protonated PsbS protein and a de-epoxidized xanthophyll. Both of these components are triggered by the lumen pH (cyan box). Reproduced from fig. 2 in [52].

and Hill coefficients [6]. As is likely to be apparent, the model contains a great many parameters—78 in all—but most of them are taken directly from the literature.

The evolution of the population of excited Chl molecules (Chl*) is modelled by chemical kinetics expressions based on the assumption that both an activated PsbS and a de-epoxidized

royalsocietypublishing.org/journal/rsob Open Biol. 9: 190043

royalsocietypublishing.org/journal/rsob Open Biol. 9: 190043

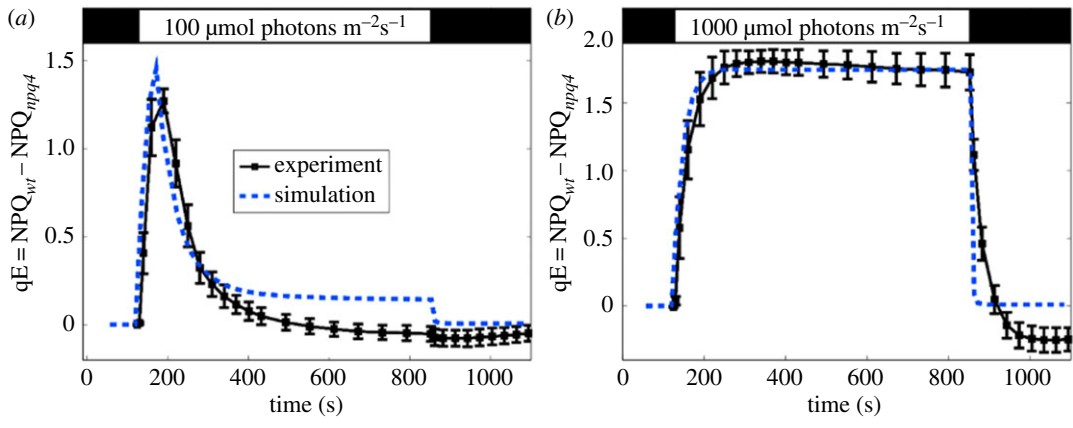

**Figure 5.** Measured (squares) and simulated (dashed lines) qE for input light intensities of (a) 100 and (b) 1000 μmol photons m$^{-2}$ s$^{-1}$. Other than light intensity, all parameters for the simulation are the same. qE is taken to be the difference in NPQ between the wild-type and *npq4* mutant lacking PsbS in order to subtract the baseline of slowly reversible NPQ. Both measured and simulated NPQ values are determined from the PAM traces. The black bar at the top indicates times when the plant is darkened, and the white bar indicates actinic light illumination. Reproduced from fig. 4 in [52].

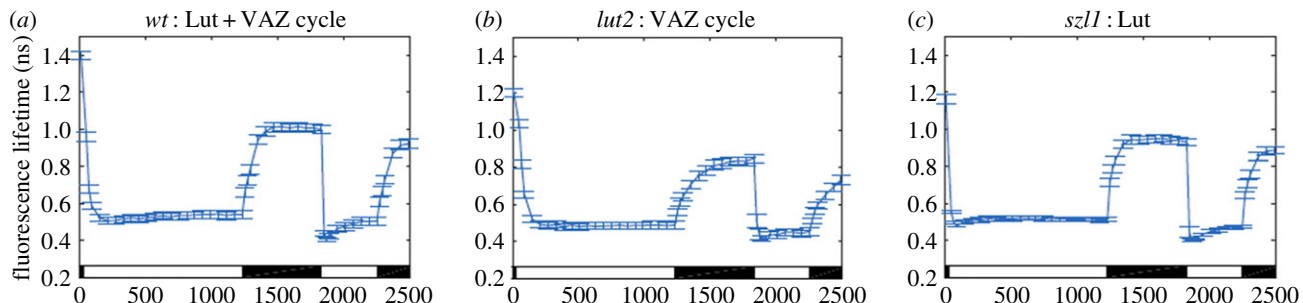

**Figure 6.** Average fluorescence lifetime traces over a two-cycle light acclimation scheme shown by the light and dark bars superimposed on the bottom of each plot for three *A. thaliana* strains. Error bars denote s.d. for $n = 20$. (a) The *wt* contains lutein and a VAZ cycle to form zeaxanthin in high light conditions. (b) The *lut2* lacks lutein and has an active VAZ cycle. (c) The *szl1* lacks zeaxanthin because of a partially blocked β-carotene biosynthesis pathway and contains more lutein than *wt*. Reproduced from fig. 2A-C in [60].

xanthophyll (e.g. zeaxanthin (Zea)) are present in the PSII membrane system. These assumptions are based on mutant studies where PsbS or VDE are absent. As an aside we note that lutein can replace Zea as the active carotenoid [30,61], but that quantitative modelling (see following section, 'A kinetic model of qE biochemistry') suggests on a per-molecule basis that Zea is about 10 times more effective in producing quenching than lutein [60]. Returning to the model, if the protonation of PsbS and VDE are assumed to be independent [62], an oversimplified but reasonable way to write the fraction of quenching sites able to dissipate Chl* excitation by qE, [Q] is

$$[Q] = F_{PsbS}[PsbS^*](|Z| + |A|), \tag{3.1}$$

with $F$[PsbS*] the fraction of PSIIs with protonated PsbS, and $|Z|$ ($|A|$) the fraction of xanthophyll binding sites that contain zeaxanthin (antheraxanthin). Figure 5 shows a comparison of the model and PAM fluorescence data for the induction of qE at two light intensities. Aside from the light intensity no other parameters are changed in the two calculations. Within the model, the turn on of qE is controlled by the time scale of Zea production, and the turn off by the decrease in protonated PsbS. The model suggests that the maximum level of qE is set by the amount of Zea, but that the quenching can be turned off relatively quickly by the deprotonation of PsbS. A second conclusion from the model is that qE has very little effect on the lumen pH suggesting that qE does not lead to a significant reduction in linear electron flow [52].

In the next section, we show how a greatly simplified version of the Zaks *et al.* model can be used to deconvolute the roles of lutein and Zea in a series of *A. thaliana* mutants.

## 3.2. A kinetic model of qE biochemistry

A simpler phenomenological model of the biochemical response to changes in light condition requires fewer free parameters and offers new insight into the relative role of different quenching mechanisms. Figure 6 shows fluorescence lifetime snapshots at varying 20–60 s time intervals over two light/dark exposure cycles for wild-type and two mutant lines. The wild-type (*wt*) contains a constant concentration of lutein and VDE that de-epoxidates violaxanthin to form zeaxanthin, via antheriaxanthin, that accumulates in response to high light. Zea is then slowly re-epoxidized in the dark, creating a 'VAZ cycle' as shown in figure 4*c*. The *lut2* mutant lacks lutein but synthesizes and accumulates excess Zea in high light [55]. The *szl1* mutant synthesizes only minimal Zea in high light, but contains excess lutein [53]. Therefore, the *lut2* mutant isolates the Zea contribution and the *szl1* mutant isolates the lutein contribution to the wild-type response. The lifetimes are used to calculate a dimensionless quenching parameter, Q, which in turn is partitioned into reversible and irreversible (or slowly reversible) components using additional qE-deficient mutants

royalsocietypublishing.org/journal/rsob   Open Biol. 9: 190043

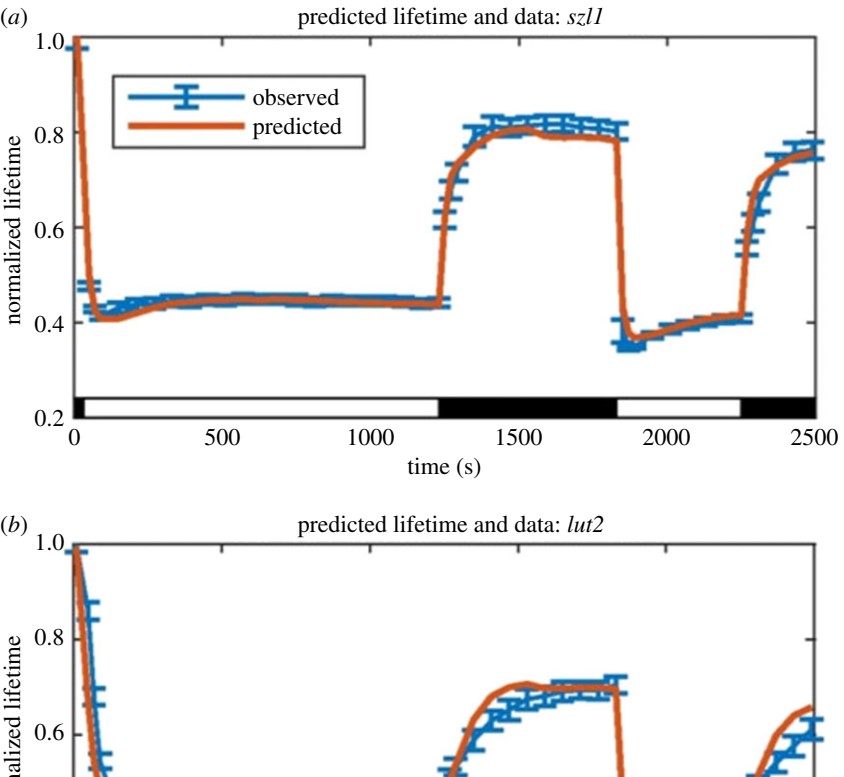

**Figure 7.** Comparison of predicted and observed lifetimes for *szl1* and *lut2*. (*a*) Normalized fluorescence lifetimes predicted by the model of reversible quenching for the *szl1* mutant (red line) lacking zeaxanthin compared with *szl1* lifetime data (blue line; with data points with error bars denoting s.d. for *n* = 20). (*b*) Normalized fluorescence lifetimes predicted by the model of reversible quenching for the *lut2* mutant (red line) lacking lutein compared with *lut2* lifetime data (blue line; with data points with error bars denoting s.d. for *n* = 20). Reproduced from fig. 4B and fig. 5B in [60].

(fig. 2 in [52]):

$$\tau \propto \phi = \frac{k_{\text{fluo}}}{k_{\text{fluo}} + k_{\text{other}} + k_{\text{quenching}}Q} \tag{3.2}$$

$$\text{and} \quad Q = Q_{\text{rev}} + Q_{\text{irr}}. \tag{3.3}$$

qE, our focus in this article, contributes to the reversible component of quenching ($Q_{\text{rev}}$). In the simplest case, quenching attributable to a single carotenoid at a constant concentration, is modelled as a two-state system,

$$\frac{\mathrm{d}}{\mathrm{d}t}Q^{\text{active}} = k^{\text{activation}}(t)Q^{\text{inactive}} - k^{\text{recovery}}Q^{\text{active}} \tag{3.4}$$

and

$$\frac{\mathrm{d}}{\mathrm{d}t}Q^{\text{inactive}} = -k^{\text{activation}}(t)Q^{\text{inactive}} + k^{\text{recovery}}Q^{\text{active}}, \tag{3.5}$$

with the time-dependent activation rate

$$k^{\text{activation}}(t) = \frac{[\text{PsbS}^*]^n}{K_{[\text{PsbS}^*]} + [\text{PsbS}^*]^n}k^{\text{activation}}, \tag{3.6}$$

where $K_{[\text{PsbS}^*]}$ and $n$ describe the effective equilibrium point and interaction coefficient of a phenomenological Hill Equation describing quenching sites' response to a time-dependent value of active [PsbS*]. To account for time-varying xanthophyll occupation, the time-independent

activation rate includes an additional phenomenological Hill equation term describing the response of quenching sites to time-dependent xanthophyll behaviour, with $K_{[\text{xanth}]}$ and $n$ defined similarly.

$$k^{\text{activation}}(t) = \frac{[\text{PsbS}^*]^n}{K_{[\text{PsbS}^*]} + [\text{PsbS}^*]^n}\frac{[\text{Xanth}]^n}{K_{[\text{Xanth}]} + [\text{Xanth}]^n}k^{\text{activation}}. \tag{3.7}$$

Note that given the observation of quenching in both the *lut2* (lacks lutein) and *szl1* mutants (minimal Zea), there are two different quenching concentrations and Hill equations—one for each xanthophyll. In addition, in the case of zeaxanthin-dependent quenching, an additional term is needed in equations (3.4) and (3.5) to account for a zeaxanthin-dependent, but pH-independent quenching, qZ [60].

The simple phenomenological description is capable of capturing the correct form of qE activation when fit to fluorescence lifetime data for the appropriate mutants (figure 7, experiment in blue, fit curve in orange). Although the fits are good in all time periods, it is appropriate to be sceptical of such fitting of data (or the resulting fit parameters). A more searching test is shown in figure 8, where the wild-type response is reconstructed from the separate contributions of lutein and Zea as extracted from the mutants. Figure 8*a* shows the reversible contribution to the quenching,

royalsocietypublishing.org/journal/rsob    Open Biol. **9**: 190043

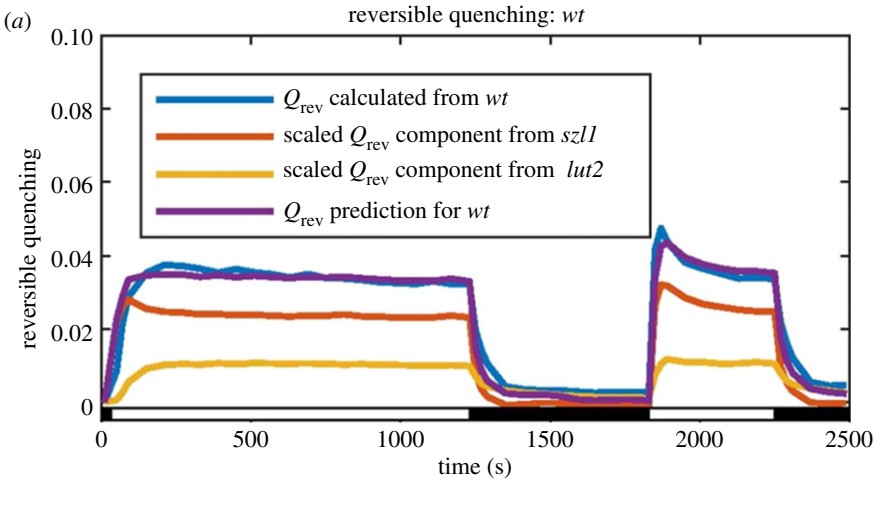

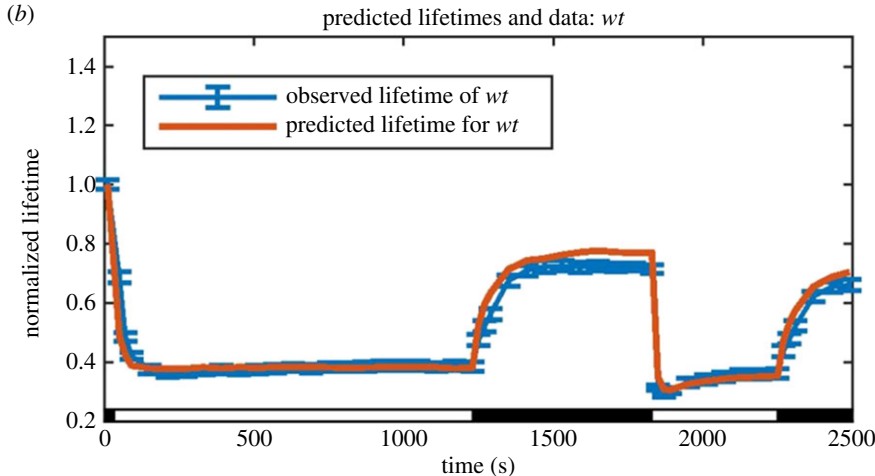

**Figure 8.** Comparison of *wt* reversible quenching calculated from lifetime data with predicted values calculated from *szl1* and *lut2* mutant lifetime data and corresponding lifetimes. (*a*) Reversible quenching calculated via equations (3.2) and (3.3) from *wt* lifetime data (blue) agrees well with the predicted reversible quenching (purple) obtained from *szl1* (red) and *lut2* (yellow) contributions via equation (3.8). (*b*) The *wt* lifetimes (red) predicted from contributions of *szl1* and *lut2* are within the error of the observed lifetimes of *wt* (blue; including error bars indicating s.d. for $n = 20$). Reproduced from fig. 6 in [60].

$Q_{\mathrm{rev}}$, for the wild-type calculated directly from the lifetime data, along with that predicted from the *lut2* and *szl1* fits using the expression

$$Q_{\mathrm{rev}}^{wt} = \alpha\left(\frac{\langle[\mathrm{Lut}]\rangle_{wt}}{\langle[\mathrm{Lut}]\rangle_{szl1}}Q_{\mathrm{rev}}^{szl1} + \frac{\langle[\mathrm{Zea}]\rangle_{wt}}{\langle[\mathrm{Zea}]\rangle_{lut2}}Q_{\mathrm{rev}}^{lut2}\right). \qquad (3.8)$$

Here $\alpha$ is a single, common scaling factor of 1.37 determined by fitting, and the lutein and Zea ratios in equation (3.8) determined from HPLC data. In figure 8*b*, the lifetimes predicted by the reconstruction are also shown to be in good agreement with the measured lifetimes. The capability of reproducing WT response with a single overall scaling factor is a promising result and suggests that the underlying phenomenological model has correctly captured the essential dynamics. The fact that the scaling factor, $\alpha$, is greater than one suggests that in combination lutein and Zea are somewhat more effective then when present singly, perhaps by alteration of binding constants. However, the finding that both zeaxanthin and lutein, can operate independently makes it unlikely that zeaxanthin serves only as an allosteric regulator as has been suggested [63–66]. Another interesting consequence of this model is that zeaxanthin is found to have a 10-fold higher capacity for quenching on a per-molecule basis than lutein.

## 4. Multiscale photophysical model of NPQ

Once the biochemical processes have activated the additional quenching pathways in response to bright light, we still struggle to understand the photophysical dynamics that underpin quenching. As we discussed above, there are a small collection of potential mechanisms that are consistent with current experimental evidence. One direction in modelling non-photochemical quenching is establishing mechanisms of quenching in isolated pigment–protein complexes, often using a combination of electronic structure and quantum dynamics simulations. While these calculations can provide detailed insight into how the specific atomic structure of pigments and the surrounding protein cavity lead to the observed dynamics, they do not represent the natural state of the light-harvesting system. The thylakoid membrane, as previously mentioned, is densely packed with pigments, and non-photochemical quenching also depends on how fast excitation moves between pigment–protein complexes. Here, we describe a multiscale model of qE in a 200 × 200 nm section of a thylakoid membrane.

Building a quantitative model of the relationship of qE to the photochemical yield ($\Phi_{\mathrm{PC}}$) requires a reconciliation of events occurring on the length and time scales of individual pigment–protein complexes with data taken on the entire

royalsocietypublishing.org/journal/rsob    Open Biol. **9**: 190043

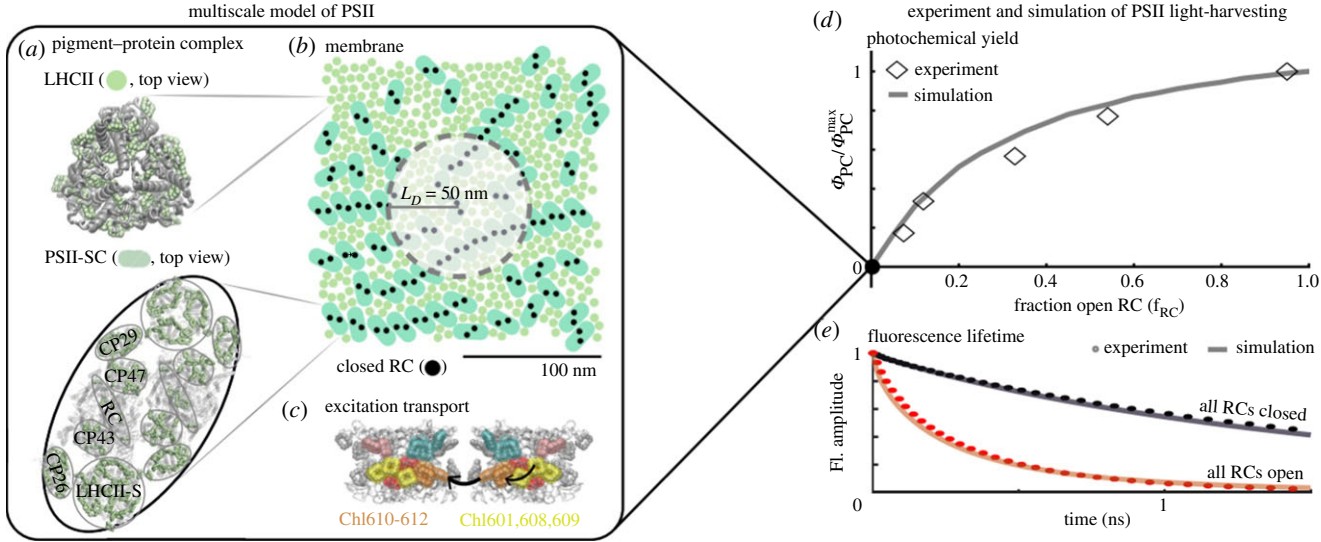

**Figure 9.** A multiscale model of PSII light harvesting reproduces experimental data on dark-acclimated leaves. (*a*) PSII is composed of two types of pigment–protein complexes: the LHCIIs and PSII-SCs. The multiscale model represents these complexes using their crystal structures [26,27]. The PSII-SC is a dimer, with each monomer containing one RC, a pair of core antenna proteins (CP43 and CP47), a pair of minor LHCs (CP26 and CP29) replaced by LHCII monomers, and a strongly bound LHCII. The pigments are indicated in light green, and the surrounding protein scaffold is in grey. (*b*) PSII harvests sunlight across the mesoscopic (approximately hundreds of nanometres) thylakoid membrane. The model arranges the crystal structures of LHCII (indicated by circles in the membrane image) and PSII-SCs (pills) into a mixed configuration. The membrane image indicates the organization of LHCII and PSII-SCs used for all simulations in this work and omits the pigment-level detail for visual clarity. The filled black circles indicate closed RCs. The radius of the shaded circle is equal to the excitation diffusion length ($L_D = 50$ nm), indicating the spatial extent of transport for an initial excitation at the centre of the circle. (*c*) Energy transfer (black arrows) is described using generalized Forster theory between domains of approximately three to four tightly coupled Chls (coloured pigments). This approach reproduces spectroscopic time scales taken on isolated LHCII and PSII-SCs [68,71,72]. (*d*) The multiscale simulation of PSII light harvesting (solid grey line) reproduces the hyperbolic dependence of the photochemical yield (diamonds) on the fraction of open RCs ($f_{RC}$) as measured by Joliot & Joliot [73] and reproduced in [74]. (*e*) Simulation (solid lines) of fluorescence lifetime measurements (dotted lines) taken on intact membranes or leaves in different states. Red indicates a state of open RCs with no qE (all RCs open), and black indicates closed RCs with no qE (all RCs closed). Reproduced from fig. 1 in [70].

functional photosynthetic membrane. qE acts on the individual pigment scale, while photochemical yield is the result of productive charge separation at all open reaction centres across the thylakoid membrane.

At the membrane scale, the intuitively named 'lake' and 'puddle' models are often used to relate the photochemical yield to the chlorophyll fluorescence yield [67]. In the puddle picture, each reaction centre has its own antenna system and excitations are able to visit, at most, a single reaction centre. By contrast, in the lake picture the reaction centres sit in a sea of antennae and an excitation has the possibility to reach multiple reaction centres. Clearly what distinguishes these two pictures is how far an excitation can travel before it is captured at a reaction centre or is dissipated by other means. However, both these limiting models neglect the finite length scale of excitation transport in the membrane. As a result, while the lake and puddle models can be useful when contemplating steady-state measurements where the excitation diffusion length is constant, they become unreliable when modelling the (de)activation of qE.

A multiscale model is required to capture the competition between quenching via qE and trapping at open reaction centres in the context of a substantial, but finite, range of excitation motion. In a series of three papers, Bennett, Amarnath and Fleming developed such a model that includes approximately 30 000 chromophores corresponding to a 200 × 200 nm patch of the PSII membrane in the presence of qE [68–70]. Figure 9 illustrates the model and shows how it captures the hyperbolic shape of oxygen evolution as a function of the fraction of open reactions centres as measured by

Joliot & Joliot [73,74], as well as the fluorescence decays of intact membranes or leaves with either all reaction centres open [75] or all reaction centres closed [50].

The multiscale model was built progressively using structural, spectroscopic and biochemical data. First, a model of the PSII supercomplex was built [68], followed by the thylakoid membrane [69] leveraging the structural model of Schneider & Geissler [41]. Finally, qE was explicitly incorporated within a membrane model of excitation transfer [70].

The membrane model provides a microscopic picture of PSII light harvesting in dim light when all RCs are open: following absorption of a photon from sunlight the excitation makes a two-dimensional random walk until an open reaction centre is reached. An open RC acts as a strong trap from which the excitation is unlikely to escape. The two-dimensional spread of excitations can be characterized by a single parameter, the excitation diffusion length, $L_D$. In the PSII antenna $L_D$ is about 50 nm (the width of the population distribution when $1/e$ of the initial excitation remains). Within $L_D$ there are likely to be many reaction centres (approx. 20). The relative spacing of reaction centres compared to the diffusion length scale explains the parabolic curve in figure 9*d*: if an excitation first encounters a closed RC, the long excitation diffusion length means that it has a high probability of reaching another reaction centre before the excitation decays by radiative or non-radiative processes. Thus, when 50% of the RCs are closed the photochemical yield is higher than 50% of the yield when all of the RCs are open.

We find that two-dimensional diffusion is also a useful model for light harvesting in the presence of qE. If the

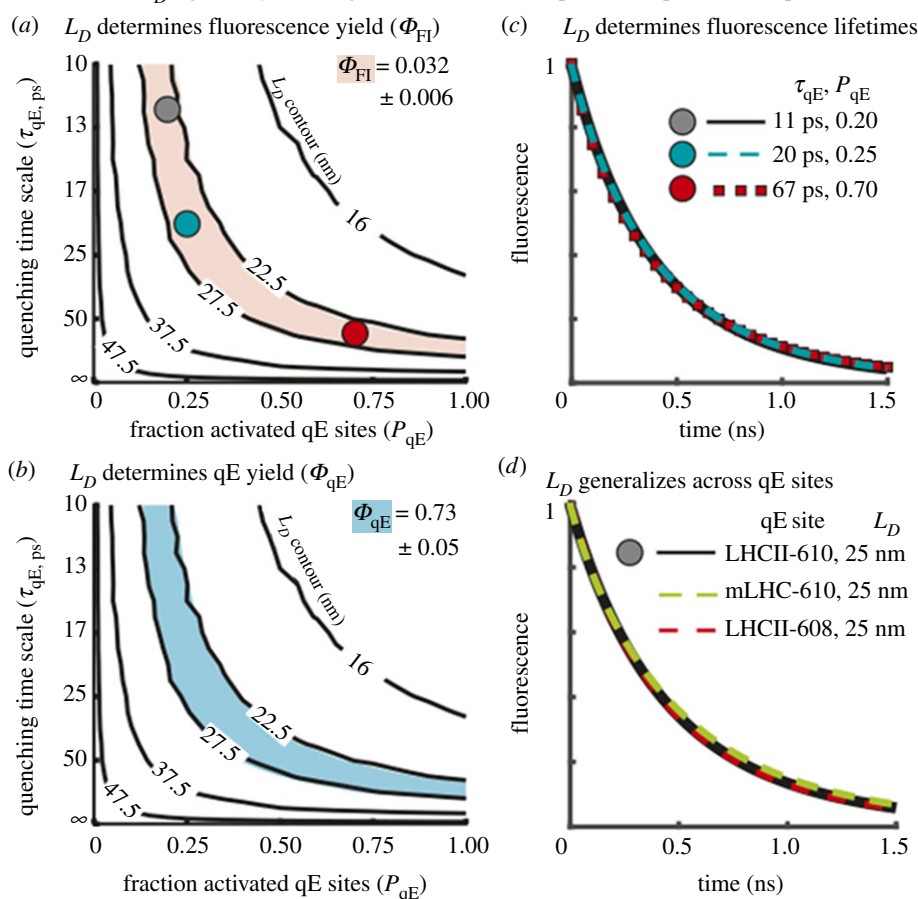

**Figure 10.** (a) Contour plot (black lines) of excitation diffusion length ($L_D$) as a function of $\tau_{qE}$ and $P_{qE}$. The red area indicates a Chl fluorescence yield ($\Phi_{FI}$) of $0.032 \pm 0.006$, consistent with the best-fit Chl fluorescence lifetime for the light-acclimated state. The coloured circles correspond to the matched fluorescence lifetimes in (c). (b) The contour plot (black lines) of excitation diffusion length ($L_D$) as a function of $\tau_{qE}$ and $P_{qE}$. The cyan area indicates a fraction of excitation quenched by qE [qE yield ($\Phi_{qE}$)] equal to $0.73 \pm 0.05$. (c) Three fluorescence lifetimes are plotted corresponding to ($\tau_{qE}$, $P_{qE}$) combinations with an excitation diffusion length of 25 nm—the matched points are shown in c. (d) Three fluorescence lifetimes are plotted corresponding to different sites of qE with combinations of ($\tau_{qE}$, $P_{qE}$) that give an excitation diffusion length of 25 nm. The black line corresponds to an LHCII-610 quenching site, which was used to generate simulation data for a–c. Simulation results using an mLHC-610 (dashed green line; $\tau_{qE} = 20$ ps, $P_{qE} = 1$) and LHCII-608 (dashed red line; $\tau_{qE} = 10$ ps, $P_{qE} = 1$) quenching site are also shown. Reproduced from fig. 2C–F in [70].

quenching rate is slow enough, we expect excitations to visit a quenching site multiple times before quenching occurs and the overall process will be well described by two-dimensional diffusion. This 'weak quenching' regime occurs when the quenching process(es) responsible for qE has an intrinsic rate less than the inverse of the 'dwell time' of an excitation on the Chl site(s) at which qE occurs. Said more simply, we should expect two-dimensional diffusion if an excitation at the quenching site is more likely to 'hop' to another group of chlorophylls than it is to be quenched by qE. The lowest-energy levels of LHCII (chlorophyll 610–612) have been suggested as a possible site for quenching. The median dwell time on these states in our model is about 3 ps, so that if the intrinsic quenching rate is, for example, $(10 \text{ ps})^{-1}$, the quenching is in the weak regime and the diffusion picture of energy flow remains valid.

qE works by decreasing the diffusion length of an excitation. If we define the qE time scale as $\tau_{qE}$ and the density of quenching sites as $\rho_{qE}$, it would seem reasonable to discuss qE in terms of these two, apparently separate, quantities. However, what the multiscale model tells us is that only the combination ($\tau_{qE}$, $\rho_{qE}$) is significant and that all combinations of $\tau_{qE}$ and $\rho_{qE}$ that accurately describe the

fluorescence decay data during qE correspond to the same value of $L_D$, the diffusion length scale. This is illustrated in figure 10 where we see that the same time-resolved fluorescence profile occurs for any combination of parameters that yields the same $L_D$. As figure 10d also shows, $L_D$ is insensitive to the choice of quenching site whether on chlorophyll 610 to 612 in LHCII or in the minor light-harvesting complexes CP26 and CP29, for example.

We can use the insights from the multiscale model to re-examine the interpretation of PAM fluorescence data, the standard method of assessing light harvesting in wild-type and mutant plants and algae. A variable excitation diffusion length influences the relationship between the fraction of open reaction centres and the photochemical yield. We found (figure 11a) an empirical relationship between the value of $L_D$ and the NPQ parameter for a range of *A. thaliana* mutants. Comparing the lake and puddle models to the multiscale model we find that the puddle model was in poor agreement with the fraction of open reaction centres across the range where qE was activated. The lake model, in contrast, provided a reasonable estimate of the fraction of open reaction centres for all values of qE. As qE activates and the excitation diffusion length decreases, however, the lake model assumes

royalsocietypublishing.org/journal/rsob    Open Biol. 9: 190043

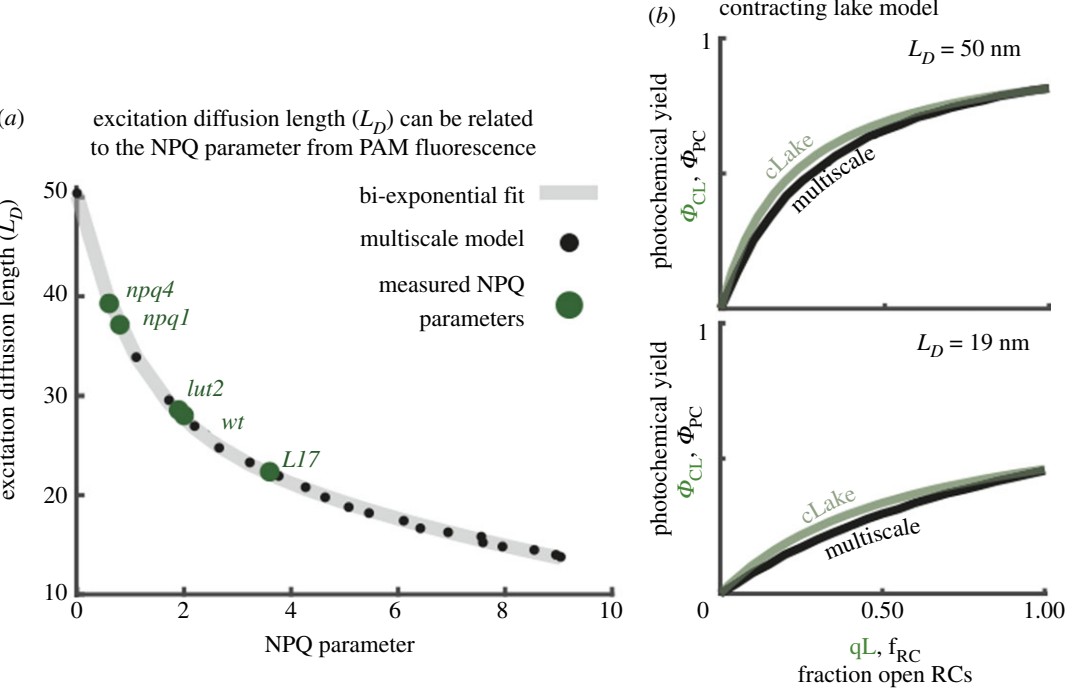

**Figure 11.** Interpreting PAM Chl fluorescence in the presence of a variable excitation diffusion length. (*a*) The excitation diffusion length is plotted as a function of the NPQ parameter extracted from Chl fluorescence simulations (black dots). A biexponential fit to these data given by $L_D = 21.44 \cdot \exp(-\text{NPQ}/1.07) + 28.76 \cdot \exp(-\text{NPQ}/12.15)$ is shown as a grey line. The green dots indicate the measured steady-state values of the NPQ parameter of several qE mutants at 1200 µmol photons m$^{-2}$ s$^{-1}$: *L17*, a PsbS overexpressor, as well as *npq1*, *lut2* and *npq4*, which are lacking Zea, lutein, and PsbS, respectively. (*b*) The photochemical yield as a function of the fraction of open RCs is plotted for both the contracting lake model (cLake; green line) and the multiscale model (black line) when $L_D = 50$ nm (Upper) and when $L_D = 19$ nm (Lower). For the case of the cLake, the *x*-axis corresponds to the qL parameter extracted from PAM measurements. Reproduced from fig. 4 in [70].

that reaction centres and qE compete for all excitations. The multiscale model, however, shows an increasing fraction of excitations that are 'dead on arrival'—they have no opportunity to reach a reaction centre before being quenched and thus cannot contribute to productive photochemistry. This constraint can be incorporated into the lake model by means of a scaling parameter, *m*, which depends on $L_D$:

$$\Phi_{\text{PC}} = m(L_d)\Phi_{\text{II}}, \tag{4.1}$$

where $\Phi_{\text{PC}}$ is the resulting photochemical yield, and $\Phi_{\text{II}}$ is the photochemical yield predicted by the lake model. The resulting 'contracting' lake model provides a good estimate of photochemical yield during qE activation and de-activation (figure 11*b*).

To sum up this rather long section, we find that the control knob activated by qE can be described by a single parameter, the excitation diffusion length, $L_D$. In other words, qE works by reducing the number of Chl molecules that can excite a given reaction centre. Capturing the competition between qE and charge separation at open RC requires modelling a region of the membrane with a diameter significantly larger than the diffusion length. Such a model enabled an empirical connection to be made between the diffusion length and the standard NPQ parameter and thus bridges the molecular and membrane length scales.

## 5. Summary and concluding comments

A prerequisite to a reliable and predictive quantitative model of rapidly reversible energy dissipation (qE) is a complete understanding of the molecular actors, their interactions and feedback loops. The experiments and models described above have provided new insight into this highly complex phenomenon. Specific photophysical mechanisms involving chlorophyll–zeaxanthin interactions have been identified, while the notion that qE works by controlling the excitation diffusion length ($L_D$) clarifies the connection between quenching sites and rates, the number of closed reaction centres, and ultimately the photochemical yield. Yet we are still some way from a complete quantitative and predictive description, let alone how the slower components of non-photochemical quenching [76] emerge from the qE phase. We need a significantly more nuanced understanding of the role of membrane morphology, the specific locations of quenching sites and their interrelation with the pH-sensing and electric-potential-sensing proteins such as PsbS [53–55,77], LHCSR [35,78,79] and LHCX [80,81]. If such a model makes accurate predictions of photosynthetic yield under natural conditions, the potential would exist for the rational design of crops with more efficient response to fluctuating light and a corresponding increase in photosynthetic productivity, perhaps by as much as 30% [9].

New experimental probes are required, such as bottom up construction of qE-like systems in liposomes or in membrane rafts. Developments in combining time and spatial resolution [82] hint at the possibility of observing the role of specific components of the antenna/supercomplex system in qE, along with the ability to directly measure the exciton diffusion length during qE. Ideally spatial resolution of less than 10 nm with single ps time resolution would be required.

The astounding range of length and time scales relevant to qE (tens of fs to minutes, and Å to micrometres) pose a major challenge to any model. We propose that systematic coarse-graining provides a powerful approach by which

this span of length and time scales can be bridged with tractable models that provide clear physical intuition. One promising avenue for future research is combining an engineering model with the coarse-grained description of excitation transport (the 'contracting lake model') established with our recent multiscale model of light harvesting. Beyond the integration of current modelling approaches lies the development of a new multiscale approach to structural changes in the thylakoid membrane, including both protein conformations and membrane organization. Current structural data are too scant to support detailed modelling, but with the continued development of NMR measurements of protein conformations [83] and single particle cryo-electron microscopy/tomography [84] a more detailed picture appears poised to emerge in the near future. Addressing this challenge will require developing new techniques for integrating experimental data and theoretical approaches to simulating protein dynamics. Combining the resulting model with a model of light-harvesting will provide a new vantage point from which to consider the underlying physics of light harvesting on a system scale.

Data accessibility. This article has no additional data.

Authors' contributions. The text describes the work of all authors: D.I.G.B. and K.A. developed the multiscale model, S.P. and C.J.S. developed and applied snapshot transient absorption spectroscopy, and J.M.M. developed the kinetic model. G.R.F. provided research direction and wrote the initial text that was expanded and edited by all authors. All authors approved the final version of the manuscript.

Competing interests. We declare we have no competing interests.

Funding. D.I.G.B. was supported by the Canadian Institute for Advanced Research through the Bio-Inspired Solar Energy Program during a portion of this work. K.A. was supported by the Simons Foundation. S.P., C.J.S., J.M.M., and G.R.F. were supported by U.S. Department of Energy, Office of Science, Basic Energy Sciences, Chemical Sciences, Geosciences, and Biosciences Division under Field Work Proposal 449B.

Acknowledgements. The experimental aspect of this work would not have been possible without the long-term collaboration of Profs K. K. Niyogi (UC Berkeley) and R. Bassi (University of Verona, Italy). During the writing of this article, G.R.F. was a visiting fellow at Magdalen College, Oxford and a visitor at the Department of Physical and Theoretical Chemistry, University of Oxford. Portions of this research used resources of the National Energy Research Computing Center, a Department of Energy user facility supported by the Office of Science contract DE-AC0205CH11231.

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
