## [Reviewer comments · Open Biology]

Review History

RSOB-19-0043.R0 (Original submission)

Review form: Reviewer 1

Recommendation

Accept with minor revision (please list in comments)

Are each of the following suitable for general readers?

- a) **Title**
Yes
- b) **Summary**
Yes
- c) **Introduction**
Yes

Is the length of the paper justified?

Yes

Should the paper be seen by a specialist statistical reviewer?

No

Is it clear how to make all supporting data available?

Yes

Is the supplementary material necessary; and if so is it adequate and clear?

Yes

Do you have any ethical concerns with this paper?

No

Comments to the Author

This is a nice overview of a difficult and often controversial area of science. There are just a few points that would help to tighten up some of the generalisations. In the Intro in para two there is a statement that a model could show that everything is understood. This only applies if that model is unique in being the only one that fully explains everything. So far most models are not unique. In Fig 1 the two npq mutants are not explained.

On a more general point the supramolecular architecture of a thylakoid membrane is rather disorganised. This means that most measurements are averaged over this disorder and this could average out key mechanistic details. This effect should be discussed.

Fig2 looks at the effects of the two proposed types of quenching. All the data presented shows that these quenching processes follow the changes in FI, but can they quantitatively account for the extent of quenching seen. This point needs a fuller discussion.

The paper presents a very clear and logical flow of the argument and will be a very useful guide for future studies.

Review form: Reviewer 2

Recommendation

Accept with minor revision (please list in comments)

Are each of the following suitable for general readers?

- a) **Title**
Yes
- b) **Summary**
Yes
- c) **Introduction**
Yes

Is the length of the paper justified?

Yes

Should the paper be seen by a specialist statistical reviewer?

No

Is it clear how to make all supporting data available?

Yes

Is the supplementary material necessary; and if so is it adequate and clear?

Not Applicable

Do you have any ethical concerns with this paper?

No

Comments to the Author

This review summarizes the ground-breaking contributions made by this group to the understanding of photosynthetic light harvesting and outlines remaining questions.

Conceptual comments

First paragraph of Introduction:

-The authors should consider replacing "can cause irreversible damage to photosynthetic proteins" with a statement like "can cause inactivation of photosynthetic proteins" because it has been shown that inactivation of D1, for example, is part of a genetic program to limit production of reactive oxygen species under highly excessive light (see Wagner et al. 2004 Science 306, 1183–1185) and probably should not be considered damage (Foyer et al. 2017 Viewing oxidative stress through the lens of oxidative signalling rather than damage. Biochemical Journal 474, 877–883; Adams et al. 2013 Photosynth Res 117, 31–44).

-NPQ is not a regulatory mechanism in the sense that it is a consequence of thermal de-excitation rather than its mechanism. Perhaps rephrase to something like "a suite of regulatory mechanisms that are often quantified from their effect on chlorophyll fluorescence as 'nonphotochemical quenching' (NPQ)."

-the statement "(qE), turns on in seconds to minutes but turns off rather slowly over a timescale of tens of minutes" is not correct as written. Under field conditions as well as under many experimental conditions, the majority of the actual qE turns off within seconds (Demmig-Adams et al. 2012 Photosynth Res 113, 75–88; Adams et al. 1999 Plant, Cell & Env 22, 125–136), and what takes longer is reconversion of zeaxanthin to violaxanthin. In addition, zeaxanthin-free leaves take longer to induce qE than zeaxanthin-containing leaves (Fig. 5 in Demmig-Adams et al. 1989 Plant Phys 90, 887–893).

Minor comments

-use single quotations marks (for British punctuation conventions) throughout

-give "*A. thaliana*" in italics

Introduction

-replace "qE is often measures" with ""qE is often measured"

-replace "plucked" with "excised" (How was the leaf maintained? With the petiole in water?)

Spectroscopic probes...

-replace "And, What..." with "And what..."

-add the reference number after "the model developed by Zaks et al."

Decision letter (RSOB-19-0043.R0)

25-Feb-2019

Dear Professor Fleming

We are pleased to inform you that your manuscript RSOB-19-0043 entitled "Models and Mechanisms of the Rapidly Reversible Regulation of Photosynthetic Light Harvesting" has been accepted by the Editor for publication in Open Biology. The reviewer(s) have recommended publication, but also suggest some minor revisions to your manuscript. Therefore, we invite you to respond to the reviewer(s)' comments and revise your manuscript.

Please submit the revised version of your manuscript within 14 days. If you do not think you will be able to meet this date please let us know immediately and we can extend this deadline for you.

- 1) A text file of the manuscript (doc, txt, rtf or tex), including the references, tables (including captions) and figure captions. Please remove any tracked changes from the text before submission. PDF files are not an accepted format for the "Main Document".
- 2) A separate electronic file of each figure (tiff, EPS or print-quality PDF preferred). The format should be produced directly from original creation package, or original software format. Please note that PowerPoint files are not accepted.
- 3) Electronic supplementary material: this should be contained in a separate file from the main text and meet our ESM criteria (see <http://royalsocietypublishing.org/instructions-authors#question5>). All supplementary materials accompanying an accepted article will be treated as in their final form. They will be published alongside the paper on the journal website and posted on the online figshare repository. Files on figshare will be made available approximately one week before the accompanying article so that the supplementary material can be attributed a unique DOI.

Online supplementary material will also carry the title and description provided during submission, so please ensure these are accurate and informative. Note that the Royal Society will not edit or typeset supplementary material and it will be hosted as provided. Please ensure that

the supplementary material includes the paper details (authors, title, journal name, article DOI). Your article DOI will be 10.1098/rsob.2016[*last 4 digits of e.g. 10.1098/rsob.20160049*].

4) A media summary: a short non-technical summary (up to 100 words) of the key findings/importance of your manuscript. Please try to write in simple English, avoid jargon, explain the importance of the topic, outline the main implications and describe why this topic is newsworthy.

Images

Data-Sharing

It is a condition of publication that data supporting your paper are made available. Data should be made available either in the electronic supplementary material or through an appropriate repository. Details of how to access data should be included in your paper. Please see <http://royalsocietypublishing.org/site/authors/policy.xhtml#question6> for more details.

Data accessibility section

Sincerely,

The Open Biology Team
<mailto:openbiology@royalsociety.org>

Reviewer(s)' Comments to Author:

Referee: 1

Comments to the Author(s)

This is a nice overview of a difficult and often controversial area of science. There are just a few points that would help to tighten up some of the generalisations. In the Intro in para two there is a statement that a model could show that everything is understood. This only applies if that model is unique in being the only one that fully explains everything. So far most models are not unique. In Fig 1 the two npq mutants are not explained.

On a more general point the supramolecular architecture of a thylakoid membrane is rather disorganised. This means that most measurements are averaged over this disorder and this could average out key mechanistic details. This effect should be discussed.

Fig2 looks at the effects of the two proposed types of quenching. All the data presented shows that these quenching processes follow the changes in Fl, but can they quantitatively account for the extent of quenching seen. This point needs a fuller discussion.

The paper presents a very clear and logical flow of the argument and will be a very useful guide for future studies.

Referee: 2

Comments to the Author(s)

This review summarizes the ground-breaking contributions made by this group to the understanding of photosynthetic light harvesting and outlines remaining questions.

Conceptual comments

First paragraph of Introduction:

-The authors should consider replacing “can cause irreversible damage to photosynthetic proteins” with a statement like “can cause inactivation of photosynthetic proteins” because it has been shown that inactivation of D1, for example, is part of a genetic program to limit production of reactive oxygen species under highly excessive light (see Wagner et al. 2004 *Science* 306, 1183–1185) and probably should not be considered damage (Foyer et al. 2017 *Viewing oxidative stress through the lens of oxidative signalling rather than damage. Biochemical Journal* 474, 877–883; Adams et al. 2013 *Photosynth Res* 117, 31–44).

-NPQ is not a regulatory mechanism in the sense that it is a consequence of thermal de-excitation rather than its mechanism. Perhaps rephrase to something like “a suite of regulatory mechanisms that are often quantified from their effect on chlorophyll fluorescence as ‘nonphotochemical quenching’ (NPQ).”

-the statement “(qE), turns on in seconds to minutes but turns off rather slowly over a timescale of tens of minutes” is not correct as written. Under field conditions as well as under many experimental conditions, the majority of the actual qE turns off within seconds (Demmig-Adams et al. 2012 *Photosynth Res* 113, 75–88; Adams et al. 1999 *Plant, Cell & Env* 22, 125–136), and what takes longer is reconversion of zeaxanthin to violaxanthin. In addition, zeaxanthin-free leaves take longer to induce qE than zeaxanthin-containing leaves (Fig. 5 in Demmig-Adams et al. 1989 *Plant Phys* 90, 887–893).

Minor comments

-use single quotation marks (for British punctuation conventions) throughout

-give “*A. thaliana*” in italics

Introduction

-replace “qE is often measures” with “qE is often measured”

-replace “plucked” with “excised” (How was the leaf maintained? With the petiole in water?)

Spectroscopic probes...

-replace “And, What...” with “And what...”

-add the reference number after “the model developed by Zaks et al.”

Author's Response to Decision Letter for (RSOB-19-0043.R0)

See Appendix A.

Decision letter (RSOB-19-0043.R1)

07-Mar-2019

Dear Professor Fleming

We are pleased to inform you that your manuscript entitled "Models and Mechanisms of the Rapidly Reversible Regulation of Photosynthetic Light Harvesting" has been accepted by the Editor for publication in Open Biology.

Sincerely,

The Open Biology Team
mailto: openbiology@royalsociety.org

Appendix A

UNIVERSITY OF CALIFORNIA, BERKELEY

BERKELEY • DAVIS • IRVINE • LOS ANGELES • MERCED • RIVERSIDE • SAN DIEGO • SAN FRANCISCO

SANTA BARBARA • SANTA CRUZ

Professor Graham R. Fleming
DEPARTMENT OF CHEMISTRY
BERKELEY, CA 94720-1460

Phone: (510) 643-3944
Fax: (510) 642-6340
Email: GRFleming@lbl.gov

March 6, 2019

Dear Editor,

We thank the reviewers for their insightful and helpful comments on our manuscript. We respond with the changes we have made below.

Yours sincerely,

Graham R. Fleming
Professor of Chemistry
University of California, Berkeley

Manuscript ID: RSOB-19-0043

Title: Models and Mechanisms of the Rapidly Reversible Regulation of Photosynthetic Light Harvesting

Author(s): Doran I. G. Bennett, Kapil Amarnath, Soomin Park, Collin J. Steen, Jonathan M. Morris, and Graham R. Fleming

Corresponding Author: Dr. Graham R. Fleming

Corresponding Author's email: grfleming@lbl.gov

Referee: 1

Comment 1: In the Intro in para two there is a statement that a model could show that everything is understood. This only applies if that model is unique in being the only one that fully explains everything. So far most models are not unique.

Response: We agree with this comment and have reworded the sentence to say:

“A model capable of quantitatively predicting the influence of qE on the kinetics of the light reactions in the presence of genetic and environmental perturbations could subsequently be incorporated into larger scale models of photosynthesis and crop yield.”

Comment 2: In Fig 1 the two npq mutants are not explained.

Response: The data for the two mutants are not necessary for this figure and we have removed them. Thanks.

Comment 3: On a more general point the supramolecular architecture of a thylakoid membrane is rather disorganised. This means that most measurements are averaged over this disorder and this could average out key mechanistic details. This effect should be discussed.

Response: We have included a short paragraph at the end of the introduction (page 6) discussing the role of disorder in the spectroscopic measurements.

Comment 4: Fig2 looks at the effects of the two proposed types of quenching. All the data presented shows that these quenching processes follow the changes in F1, but can they quantitatively account for the extent of quenching seen. This point needs a fuller discussion.

Response: We added the following paragraph at the end of the ‘Spectroscopic Probes’ section:

“It is clearly important to estimate if energy and charge transfer can account for all of qE. However, at least two factors complicate such an estimate. First, the influence of exciton annihilation in the snapshot transient absorption measurements needs to be accurately modeled via a multiscale approach. Second, accurate values for the extinction coefficients of Zea S₁ and Zea^{•+} *in vivo* are required. Particularly in the latter case there is a great deal of uncertainty about the value in part because of proton loss from the radical (Focsan et al *Archives Biochem Biophys* 2015, Park et al *JPCL* 2017).”

Referee: 2

Conceptual comments

Comment 1: The authors should consider replacing “can cause irreversible damage to photosynthetic proteins” with a statement like “can cause inactivation of photosynthetic proteins” because it has been shown that inactivation of D1, for example, is part of a genetic program to limit production of reactive oxygen species under highly excessive light (see Wagner et al. 2004 *Science* 306, 1183–1185) and probably should not be considered damage (Foyer et al. 2017 *Viewing oxidative stress through the lens of oxidative signalling rather than damage. Biochemical Journal* 474, 877–883; Adams et al. 2013 *Photosynth Res* 117, 31-44).

Response: We thank the reviewer for suggesting this clarification. We have reworded the first paragraph along the lines suggested. We have also added a reference to Wagner et al 2004.

Comment 2: NPQ is not a regulatory mechanism in the sense that it is a consequence of thermal de-excitation rather than its mechanism. Perhaps rephrase to something like “a

suite of regulatory mechanisms that are often quantified from their effect on chlorophyll fluorescence as ‘nonphotochemical quenching’ (NPQ).”

Response: Again we agree that the reviewer’s wording is more precise than our original text. We have used the suggested rephrasing.

Comment 3: the statement “(qE), turns on in seconds to minutes but turns off rather slowly over a timescale of tens of minutes” is not correct as written. Under field conditions as well as under many experimental conditions, the majority of the actual qE turns off within seconds (Demmig-Adams et al. 2012 Photosynth Res 113, 75-88; Adams et al. 1999 Plant, Cell & Env 22, 125-136), and what takes longer is reconversion of zeaxanthin to violaxanthin. In addition, zeaxanthin-free leaves take longer to induce qE than zeaxanthin-containing leaves (Fig. 5 in Demmig-Adams et al. 1989 Plant Phys 90, 887-893).

Response: Thank you for pointing this out. We have corrected our statement about the timescales as suggested by the reviewer and have refocused the sentence on the significance of qE in the presence of fluctuating light.

Comment 4:

Minor comments:

- use single quotations marks (for British punctuation conventions) throughout
- give “*A. thaliana*” in italics

Introduction

- replace “qE is often measures” with ““qE is often measured”
- replace “plucked” with ”excised” (How was the leaf maintained? With the petiole in water?)

Spectroscopic probes...

- replace “And, What...” with “And what...”
- add the reference number after “the model developed by Zaks et al.”

Response: We have applied all of these changes in the revised manuscript and specified that the petiole was immersed in water during measurements.